# Thermal Catalytic-Cracking Low-Density Polyethylene Waste by Metakaolin-Based Geopolymer NaA Microsphere

**DOI:** 10.3390/molecules27082557

**Published:** 2022-04-15

**Authors:** Shanshan Tang, Yan He, Xingfa Deng, Xuemin Cui

**Affiliations:** Guangxi Key Lab of Petrochemical Resource Processing and Process Intensification Technology, School of Chemistry and Chemical Engineering, Guangxi University, Nanning 530004, China; 1914391049@st.gxu.edu.cn (S.T.); 20130017@gxu.edu.cn (Y.H.); 1814304005@st.gxu.edu.cn (X.D.)

**Keywords:** geopolymers, NaA molecular sieve, catalysts, plastic waste, fuel oil

## Abstract

Metakaolin-based geopolymer microspheres (MGM) with hierarchical pore structures were prepared by suspension dispersion method in dimethicone at 80 °C. The hydrothermal modification of MGM was carried out at a lower temperature of 80 °C, and a NaA molecular sieve converted from metakaolin-based geopolymer (NMGM) with good crystal structure was prepared and applied in thermal catalytic cracking of low-density polyethylene (LDPE) reaction. The one-pot two-stage thermal catalytic cracking of LDPE was carried out in a 100 mL micro-autoclave under normal pressure. In this work, the optimal proportions and optimal reaction conditions of catalysts for NMGM thermal catalytic cracking of LDPE waste to fuel oil were investigated. The NMGM catalyst showed high selectivity to the liquid product of thermal catalytic cracking of waste LDPE. Under the reaction conditions of reaction time of 1 h and reaction temperature of 400 °C, the liquid-phase yield of thermal catalytic cracking of LDPE reached a high of 88.45%, of which the content of gasoline components was 10.14% and the content of diesel components was 80.97%.

## 1. Introduction

Plastics are durable and cheap, so they have become a common item in thousands of households, which greatly facilitates human life. It was once hailed as one of the greatest inventions of the 20th century. Over time, plastic has gone from a lauded invention to one of the biggest enemies of environmental pollution. Although the pace of technological development continues to advance, the problem of plastic pollution has not yet been fundamentally solved. Microplastics, which have been widely criticized in recent years, refer to plastic particles smaller than 5 mm that cannot be degraded in a short period of time and thus remain in the environment, and are difficult to be detected by the naked eye [1,2]. Microplastics have been detected in the sparsely populated Arctic [3,4]. Microplastics enter the diet of marine organisms, accumulate in the biological chain, and then enter the human table, endangering human health [5].

Landfilling plastics in traditional plastic-disposal methods will bring about secondary pollution problems. Plastics have a high calorific value (CV) that matches conventional fuels such as gasoline, kerosene, diesel, etc., so incinerating plastics to generate electricity, steam and heat is also a major way of recycling. However, its economic value has been questioned and the issues of unstable combustion and the emission of toxic pollutants such as dioxins and furans have not been resolved [6,7]. The above two methods will bring about the problem of secondary pollution, but most developing countries are still using these two types of traditional treatment solutions [8]. At present, researchers focus on two solutions to plastic pollution: one is research on degradable plastics, and the other is research on cracking plastics into high-value-added products [9,10,11,12]. The research and development of degradable plastics is considered to solve the problem of plastic pollution from the root cause, but degradable plastics are still inferior to traditional plastics in terms of finished-product quality and performance. Degradable plastics have a large gap compared with traditional plastics, with good performance in terms of tensile strength and elongation at break, and the cost is much higher than traditional plastics [13]. As a degradable plastic as a packaging material, its biodegradability needs to be improved, and it cannot degrade quickly and completely after completing the mission of packaging, and the residue is difficult to degrade, causing land pollution [14]. Therefore, there are a series of scientific and technological problems in the process of optimizing the performance of degradable plastics, which cannot fully replace traditional plastic products.

It is currently popular to recycle plastics into gaseous and liquid products with high added value [15,16,17]. Methods of gasifying plastic waste into syngas have been studied, but their recycling is expensive [18]. Cracking plastic wastes into fuel (gasoline, diesel) is convenient for transportation and has practical significance in rapidly rising oil prices. The cracking method can be divided into pyrolysis, catalytic cracking, and a method combining pyrolysis and catalytic cracking [19]. The pyrolysis reaction needs to be carried out at a high temperature of 500 to 900 °C; the operating temperature is high, and the heat demand increases sharply because the pyrolysis is an endothermic reaction [20]. The carbon-number distribution of the product after pyrolysis is wide, and further processing is required to improve the quality of the oil in the product. The catalytic-cracking method uses a catalyst, which can reduce the activation energy required for the reaction, so that the plastic is cracked at a lower reaction temperature [21]. The use of suitable catalysts can narrow the distribution of reaction products, improve the selectivity of target products, and reduce subsequent separation and processing procedures.

The combined method of thermal cracking and catalytic cracking can be divided into catalytic pyrolysis, thermal cracking-catalytic cracking, and catalytic cracking-catalytic upgrading [22]. The catalyst in the catalytic-pyrolysis process is mixed with a plastic sample in a batch reactor. The disadvantage of this process is the high tendency to form coke on the catalyst surface, which reduces catalyst efficiency over time and results in high residue yields. In addition to this, the separation of the residue from the catalyst at the end of the experiment was difficult. Catalytic cracking-catalytic upgrading means adding a small amount of catalyst in the pyrolysis stage of the two-step method, thereby shortening the cracking time and reducing the cracking temperature [23]. However, its shortcomings are also more obvious, the amount of catalyst is increased, and there is no economic benefit. The difference between the thermal cracking-catalytic cracking upgrading conversion method and the catalytic cracking-catalytic upgrading conversion method is that there is no catalyst involved in the thermal cracking process [24].

Molten plastics have high viscosity and low thermal conductivity, which will cause a series of heat- and mass-transfer problems. Therefore, many scholars have carried out research on the selection of plastic-cracking reactors, so as to improve the mass transfer and heat transfer of the reaction process. Kaminsky et al. [25] developed a process consisting mainly of a fluidized-bed reactor, ensuring a uniform temperature throughout the reactor. Fluidized beds can be operated with a continuous plastic feed, which is beneficial for scaling up the process, but when operating conditions are out of range, molten plastic-coated particles can agglomerate and bed deflow occurs [26]. The helical kiln consists of a tubular reactor and a screw conveyor. The residence time of the polymer can be controlled by changing the speed of the screw, and the heat-transfer rate and pyrolysis temperature can be well controlled [27]. The gas-solid contact in the spouted-bed reactor is violent, and it can be used to treat irregular particles, fine particles, and other materials that are difficult to deal with by other methods of gas-solid contact [28]. Fixed-bed reactors are usually operated in batch mode during the pyrolysis of waste plastics [29]. Due to the poor heat- and mass-transfer efficiency of the molten plastic, the pyrolysis products initially formed from the plastic in the fixed-bed reactor are usually passed into another reactor for further cracking under the purging of an inert gas stream. Because batch or semi-batch reactor parameters are easy to control, they are the best reactors for obtaining high liquid yields, and their main disadvantage is the tendency to form coke on the outer surface of the catalyst, thereby reducing the overall yield of liquid product [30].

In this work, the method of separating the catalyst from the cracking raw materials is adopted, and the low-density polyethylene at the bottom is heated to vaporize it through the catalyst in the upper layer, and the thermal cracking and catalytic cracking are combined: A one-pot two-stage reaction. Compared with the traditional thermal cracking-catalytic upgrading method, this method is more convenient and saves on equipment and equipment cost. This is referred to herein simply as thermal catalytic cracking.

The key to the thermal catalytic cracking-catalytic pyrolysis process is the catalyst. Due to the high activity of solid acid catalysts in the cracking of plastic wastes into fuel oil, researchers are currently focusing on this type of catalysts [31,32]. However, its high-strength acidity can easily lead to the deactivation of the catalyst. Zeolite molecular sieves with high Si/Al ratio, such as ZSM-5 [33,34], HZSM-5 [24] and Hβ [35], are all high-strength acid catalysts, which are expensive to prepare and require high-precision preparation technology. The degradation effect of different catalysts on plastics has been investigated by many researchers and examined using kinetic parameters determined by different models. Khan et al. [36] used a commercial LZ-Y52 molecular-sieve catalyst to react at 370 °C for 60 min, and the oil yield exceeded 40%. Nisar et al. [37] calculated the kinetic parameters and found that the use of catalysts can reduce the activation energy of the reaction. Aguado et al. [38] found that mesoporous catalysts with high accessibility of plastic molecules and catalysts with small crystal size (high specific surface area) are expected to improve the catalytic activity of plastic cracking.

Geopolymer is a three-dimensional network gel with amorphous and quasi-crystalline characteristics, which is formed by the polymerization of silicon-oxygen tetrahedron and aluminum-oxygen tetrahedron, which is similar to zeolite in chemical composition [39,40,41]. The production-energy consumption of geopolymer is low, equivalent to only 30% of the energy consumption of cement production. Geopolymer properties and formation methods are similar to natural zeolites. The zeolite molecular-sieve microspheres prepared from geopolymers have suitable acidity and controllable acidity. Microspheres with a hierarchical pore structure can be prepared by adding a suitable foaming agent, which provides abundant acidic reaction sites and channels for the catalytic cracking of plastics [42]. Alkali-excited geopolymers have high mechanical strength, so recovery after the reaction is complete is convenient.

At present, commercial catalysts still have problems such as low selectivity of oil products and low quality of oil products, high cost of catalyst preparation and easy deactivation, and high energy consumption caused by high cracking temperature. The problem of environmental pollution caused by plastics is becoming more and more serious, and the price of crude oil is rising. Therefore, it is of practical significance to develop an inexpensive plastic-cracking catalyst with high catalytic activity, so as to produce high value-added oil products with high selectivity.

In this work, metakaolin was used as raw material, sodium hydroxide was used as alkaline activator, and 0.1wt% H_2_O_2_ and 0.05wt% K_12_ were added to prepare geopolymer microspheres with hierarchical pore structure and low Si/Al ratio. It was hydrothermally treated and successfully converted into a metakaolin-based NaA molecular sieve (NMGM), which was used in the thermal catalytic cracking of urban solid plastic LDPE to make fuel oil. The changes of NMGM catalyst-preparation conditions and reaction conditions on the yield and hydrocarbon composition of liquid-phase products from thermal catalytic cracking of waste LDPE were investigated.

## 2. Results and Discussion

### 2.1. Physicochemical Properties

The Na_2_O/SiO_2_ of the catalyst was varied by varying the amount of NaOH added during the preparation of the NMGM catalyst. The prepared microspheres were all hydrothermally heated at 80 °C for 1 d. From the SEM test at 100 μm, the microspheres remained spherical after hydrothermal treatment (Figure 1b,d,f,h). When Na_2_O/SiO_2_ was 0.2, the molecular-sieve grains obtained by the transformation of geopolymers could not be found (Figure 1a). When the dosage of NaOH was increased to Na_2_O/SiO_2_ of 0.4, a few polyhedral grains with a diameter of about 1 μm were observed on the surface of the microspheres (Figure 1c). When the Na_2_O/SiO_2_ ratio was 0.8, a large number of cubic molecular-sieve grains with side lengths of about 0.68 μm were formed on the surface of the microspheres, and many macropores could be observed on the surface (Figure 1e). When Na_2_O/SiO_2_ was 1.2, the amount of alkali activator reached its peak, and the surface of the microspheres was denser than the first three groups. It could be observed that the molecular sieves have different grain sizes and rough edges (Figure 1g).

From Figure 1i, there were ten main characteristic peaks in the standard card spectrum of the NaA molecular sieve. The 2θ of ten main characteristic peaks were the diffraction peaks at 7.18°, 10.16°, 12.45°, 16.09°, 21.65°, 23.96°, 26.09°, 27.09°, 29.92°, and 34.17°, respectively. With the increase in alkali activator, the crystallinity of NMGM first increased and then decreased. The NaA molecular sieve with higher crystallinity could be obtained by adding an appropriate amount of NaOH. When the amount of alkali activator was insufficient, the metakaolin could not be fully excited, and it mainly presented an amorphous dispersion peak between 20° and 30° of 2θ. If the alkali activator was used in excess, due to the influence of the high-alkaline environment, the formed NaA molecular-sieve crystal grained with complete crystal form would be dissolved again [43]. In this work, a Na_2_O/SiO_2_ ratio of 0.8 was chosen as the base activator dosage for the preparation of NMGM catalysts.

The XRD test showed that the crystallinity of metakaolin-based geopolymer converted into NaA molecular sieve was affected by hydrothermal time. When the hydrothermal time was 1 d, the XRD diffraction peaks would be transformed from the initial amorphous diffraction peaks to the crystalline diffraction peaks of NaA molecular sieves. Among them, when the hydrothermal time was 2 d, the diffraction peak intensity was the strongest and the crystallinity was the highest. When the hydrothermal time was extended to 3 d, the intensity of diffraction peaks decreased instead (Figure 2a). The in situ transformation of geopolymers into molecular sieves could be divided into nucleation and growth stages. The alkalinity in the hydrothermal solvent reached the maximum when the conversion rate of molecular sieves reached the maximum. At this time, if the hydrothermal treatment was not stopped, part of the formed molecular sieve would dissolve and collapse. Therefore, the long-term hydrothermal treatment would cause the intensity of the diffraction peak to weaken.

The BET test of the microspheres subjected to different hydrothermal times clearly showed that the NMGM catalyst was a porous structure dominated by mesopores and macropores from the pore-size-distribution map (Figure 2b). The presence of catalyst mesopores plays a positive role in plastic cracking, and Sakata et al. [44] claim that KFS-16 is a pure silica mesoporous material without acid sites. However, it can crack polyethylene as quickly as silica-alumina, thereby producing more liquid. This is because mesopores act as sites for long-term storage of free-radical species, so abundant free radicals may accelerate plastic degradation. When the hydrothermal time was 1 d, the specific surface area of NMGM reached the maximum, which was 38.49 m^2^/g (Figure 2c). At this time, the NaA molecular sieve had experienced a period of growth, and the larger NaA molecular-sieve crystals grew in a staggered manner, resulting in numerous pores. The hydrothermal time continued to extend to 2 d, and the specific surface area decreased to 23.11 m^2^/g. At this time, the growth of NaA molecular-sieve crystals reached saturation, and the formation of excessive molecular-sieve crystals blocked some of the pores, resulting in a decrease in the specific surface area. When the hydrothermal time was 3 d, some of the formed NaA molecular-sieve grains were dissolved and collapsed under strong alkaline conditions, resulting in blockage and collapse of some of the pores of NMGM, and the specific surface area and pore size were reduced. The increase in specific surface area and the existence of mesopores had a positive effect on the thermal catalytic cracking of LDPE. The NMGM catalyst prepared in this work selects 1 d as the optimal hydrothermal modification time for the conversion of metakaolin-based geopolymers into NaA molecular sieves.

NH_3_-TPD tests were performed on NMGM that were not hydrothermally and hydrothermally modified for 1 d at 80 °C (Figure 3). The changes in acid properties of NMGM after hydrothermal modification were significant, and the total acid content increased from 45.01 µmol/g to 951.09 µmol/g. The total acid content and pore structure of the catalyst determine its catalytic activity in acid-catalyzed reactions [31]. After hydrothermal treatment for 1 d, the NMGM catalyst was converted from an amorphous structure to a NaA molecular sieve, resulting in a significant increase in the total acid content and abundant pores, thereby improving the reactivity of the catalyst. The total acid content of the catalyst was related to the specific surface area of the catalyst and the density of acid sites on the catalyst surface.

After hydrothermal treatment for 1 d, NMGM had three desorption peaks: the low-temperature peak corresponded to the weak acid site, the high-temperature peak corresponded to the strong-acid site, and the middle peak was the medium strong-acid site. After hydrothermal modification, the strong-acid center of NMGM did not shift greatly compared with that before hydrothermal modification, and the overall acidic-reaction site was composed of weak acid and medium strong acid, which is a catalyst of non-high-strength acid type. Sakata et al. [44] explored the effect of catalyst acidity on the distribution of HDPE pyrolysis products. The results of the acidity-strength test of the catalyst showed that SA-1 > ZSM-5 > SA-2. The results showed that SA-2 catalyst with lower acidity was observed to produce the highest amount of liquid oil, and ZSM-5 with strong-acid sites tended to produce more gaseous products with very low liquid yield. Appropriate acidity is beneficial to inhibit excessive cracking of liquid products and prepare liquid products with high selectivity. 

### 2.2. Thermal Catalytic Cracking of Waste LDPE

The molten LDPE has strong viscosity. If the waste LDPE plastic was directly mixed with the catalyst, it would easily adhere to the surface and block the catalyst pores, increasing the occurrence of coking reaction. In this experiment, the solid-plastic waste LDPE raw materials were first pyrolyzed and vaporized at the bottom of the quartz cup glass, and then catalytically cracked with the NMGM catalyst on the upper layer of quartz wool under the purging of N_2_.

Used NMGM before and after hydrothermal modification at 80 °C for 1 d, the results of catalytic thermal cracking of waste LDPE at 400 °C for 1 h are shown in Figure 4. The yield of liquid oil in the cracked product increased from 23.51 wt% to 88.45 wt%, and the solid yield decreased from 38.59 wt% to 3.48 wt% (Figure 4a). The yield of aromatic products in the liquid fraction increased from 1.18% to 6.79%. After hydrothermal treatment, the content of hydroxyl groups in the pores of the NMGM microspheres and on the surface of the microspheres decreased, so the hydroxyl groups involved in the replacement reaction decreased, and the yield of alcohol products decreased significantly, from 47.88% to 20.33% (Figure 4b).

Figure 4d shows the FTIR test results of the liquid-phase products after NMGM cracking waste LDPE before and after hydrothermal treatment. The stretching vibration peaks of the C-H bond at 2925.51 cm^−1^ and the shear and bending vibration peaks of the C-H bond at 1469.87 cm^−1^ confirmed the existence of alkanes in the liquid product, and their peak intensity was enhanced. In the infrared test results of the liquid-phase product after hydrothermal NMGM-cracking waste LDPE, the strength of the stretching vibration peak of the C=C bond at 1639.46 cm^−1^ and the C-H bending vibration peak at 967.74 cm^−1^ of the olefin were weakened. The appearance of the C-H bending vibration peak at 721.69 cm^−1^ indicates the presence of olefinic and aromatic compounds in the product.

Figure 5 shows when the reaction time of 1 h at 350 °C, 400 °C and 450 °C in a 100 mL micro-autoclave reactor system with a feed/catalyst ratio of 5, the activity test results of NMGM catalyst for thermal catalytic cracking of LDPE plastic into fuel oil.

As the reaction temperature increased from 350 °C to 450 °C, the yield of solid components gradually decreased. The yield of liquid-oil products was the highest at 400 °C, the yield of gas-phase components was the lowest at this time, and the yield of gas-phase products reached its highest at 450 °C. A low reaction temperature could not fully crack LDPE, and the increase in temperature had a positive effect on the formation of gas-phase products (Figure 5a). Onwudili et al. studied the effect of temperature and residence time on the degradation of LDPE. It has been observed that long residence times and excessively high reaction temperatures provide opportunities for secondary reactions and cracking of oil to produce gas and char [45].

The GC-MS test of the liquid oil of NMGM thermal catalytic cracking of LDPE showed that the main hydrocarbons were alkanes and alcohols, and the olefins and aromatics were relatively few. Among them, the octane number of aromatic hydrocarbons was higher, so they were good components of gasoline. If the reaction temperature was too high, it was not conducive to the formation of aromatic hydrocarbon products, and the aromatic hydrocarbon products were the highest at 400 °C, reaching 6.79% (Figure 5b). The content of gasoline components (C_5_-C_12_) was the highest when the reaction temperature was 450 °C, and the total content of gasoline and diesel components (C_5_-C_22_) reached the highest value of 91.10% at 400 °C. Among them, the decrease in diesel components (C_13_-C_22_) at 450 °C was presumed to be due to the higher temperature promoting the further cracking of diesel components (Figure 5c).

The FTIR test results of the liquid-phase products are shown in Figure 5d. The stretching vibration peak of the alkane C-H bond at 2925.51 cm^−1^ and the shear and bending vibration peak of C-H bond at 1469.87 cm^−1^ increased and then weakened after the reaction temperature increased. This was corroborated with the GC-MS test results in Figure 4b. The olefins in the liquid phase were the stretching vibration peak of C=C bond at 1639.46 cm^−1^ and the C-H bending vibration peak at 967.74 cm^−1^ gradually weakened. The C-H bending vibration peak at 721.69 cm^−1^ indicates the presence of olefin and aromatic compounds in the product. 3440.12 cm^−1^ was the bending vibration peak of alcohols.

Figure 6 shows the reaction temperatures of 0.5 h, 1 h, 2 h, and 4 h at 400 °C in a 100 mL micro-autoclave reactor system with a feed/catalyst ratio of 5, and the activity test results of the NMGM catalyst for thermal catalytic cracking of waste LDPE plastics into fuel oil. 

The longer the reaction time, the higher yield of gas-phase products. The highest yield of gas-phase products was 48.22% under the reaction time of 4 h, and the highest solid yield was 30.50% under the reaction time of 0.5 h. The yield of liquid oil was up to 88.45% under the reaction conditions of 1 h (Figure 6a).

When the reaction time was extended to 4 h, the content of alcohol compounds in the liquid product reached its highest, 61.79%. The reason was deduced that the surface of the base-excited NMGM catalyst prepared by geopolymer contained abundant hydroxyl groups, the reaction time was long, and the reaction products tended to generate relatively stable alcohol compounds in the reaction system. The content of olefins in the liquid oil gradually decreased with the prolongation of the reaction time, and the olefin products were unstable, and tended to generate more stable alcohols and alkanes in the progress of the reaction (Figure 6b). When the reaction time is 1 h, the reaction had the best selectivity to diesel components, and the content of diesel components accounted for 80.97% of the liquid oil (Figure 6c).

## 3. Materials and Methods

### 3.1. Catalyst Preparation and Thermal Catalytic-Cracking Procedure

The preparation of NMGM catalyst can be seen in Figure 7a. First, a certain amount of metakaolin was taken, an appropriate dosage of NaOH solution of 11 mol/L was added according to the required sodium-silicon ratio (Na_2_O/SiO_2_), and the amount of distilled water added was controlled according to the water-sodium ratio (H_2_O/Na_2_O) of 20. The above slurry was stirred at a speed of 1000 r/min for 5 min under a mechanical stirring paddle to a uniform state. Then, 0.05 wt% H_2_O_2_ and 0.1 wt% K_12_ (sodium dodecyl sulfate) were added; the purpose was to increase the porosity in the geopolymer slurry, and the slurry was stirred at 1000 r/min for 2 min to foam status.

In this experiment, the suspension-dispersion polymerization method was used to prepare the geopolymer-catalyst microspheres, and the prepared alkali-excited geopolymer slurry was sheared into small segments by the shearing force during the rotating operation of the stirring paddle, then dispersed into spheres under the action of surface tension of silicone oil. The prepared slurry was put into a 20 mL syringe and injected into simethicone at 80 °C. The speed of the stirring paddle was adjusted to 600 r/min, and the microspheres were rapidly solidified and formed. The microspheres were placed in an oven at 60 °C for 12 h in dimethyl silicone oil to make them completely cured.

The cured microspheres were separated from the silicone oil with a rotary-vane vacuum pump. The filtrate was dimethyl silicone oil, which could be collected for recycling. The separated microspheres were washed three times with hot distilled water to remove the dimethyl silicone oil remaining on the surface of the microspheres and the inner pores. Next, the cleaned microspheres were dried in an oven at 60 °C, and then calcined at 450 °C for 6 h to remove residual silicone oil. After the temperature of the microspheres dropped to room temperature, the microspheres were placed in a 100 mL conical flask, 50 mL of distilled water was added, and the solution was heated at 80 °C for 1 d. Finally, the microspheres were dried in an electric blast-drying oven at 60 °C.

In this work, a two-stage method was adopted. The reactor was lined with a quartz glass cup (254 mm in height and 21 mm in inner diameter); LDPE was placed at the bottom, the middle layer was quartz wool, and the catalyst was placed on the upper layer of quartz wool. The LDPE was heated and vaporized at 400 °C and reacted through the upper catalyst, as shown in Figure 7b.

The quartz glass was placed in a programmable and temperature-controlled 100 mL micro-autoclave, and the four knobs were tightened above the reactor in a clockwise direction to ensure that the inside of the reactor was airtight. Before the heating program of the reaction kettle started, N_2_ was fed with a flow rate of 100 mL/min, and the air was purged for 10 min to remove the air in the reactor. Then, the valves on both sides of the reaction kettle were tightened so that the kettle was in a closed state filled with N_2_. The K-type WRNT-042 m open-probe thermocouple produced by ELECALL, which was equipped with the reactor, was placed near the center of the reactor to monitor the real-time temperature in the reactor at all times. The heating rate was set as 4 °C/min. When the temperature in the reaction kettle reached the reaction temperature, the valves on both sides of the reaction kettle were slowly twisted, and the flow meter was adjusted to control the flow rate of N_2_ at 50 mL/min. Driven by nitrogen, the product outlet was connected to a U-shaped tube placed in a constant-temperature circulating condensing pool at 0 °C, and the condensed-liquid products were recovered. The uncondensed product was connected to the gas outlet, and the gas was collected with a 3L aluminum-foil air bag with single valve from Beekman Bio. After the reaction, the reaction kettle was cooled down to room temperature by natural cooling before the next set of experiments could be carried out.

The total yield of LDPE degradation (wt%) and the conversion rate of liquid, gas, coke, or residue (wt%) were calculated as follows:(1)$$\mathrm{Total}\;\mathrm{Yieled}\left(T\right)=\frac{\mathrm{Mp}-\mathrm{Mr}}{\mathrm{Mp}}\times 100\%$$
(2)$$\mathrm{Liquid}\;\mathrm{Yieled}\left(L\right)=\frac{\mathrm{Ml}}{\mathrm{Mp}}\times 100\%$$
(3)$$\mathrm{Gas}\;\mathrm{Yieled}\left(G\right)=\frac{\mathrm{Mp}-\mathrm{Ml}-\mathrm{Mw}-\mathrm{Mc}}{\mathrm{Mp}}\times 100\%$$
(4)$$\mathrm{Solid}\;\mathrm{Yieled}\left(S\right)=\frac{\mathrm{Mw}}{\mathrm{Mp}}\times 100\%$$

In the above formulas, Mo represents the mass of the MGM catalyst added before the reaction, Mp is the mass of LDPE, Mc is the mass of coke on the catalyst after the reaction, Mw is the mass of solid in the quartz tube after the reaction, and Ml is the mass of the liquid-phase product in the U-shaped tube after the reaction. According to the law of conservation of mass, the gas-product conversion rate was determined by subtracting the yield of the liquid-phase products and coke or residue.

### 3.2. Materials and Test Methods

The waste LDPE used in this experiment comes from plastic waste bottles, and the results obtained from the test showed that its density is 0.921 g/cm^3^. The melt mass-flow rate (mass of polymer extruded through a 1 mm die after 10 min use at 190 °C with a nominal load of 2.16 kg) was 2.00 g/10 min. The collected waste LDPE plastic bottles were cleaned, cut into small squares of 1 mm × 1 mm, and dried in a blast-drying oven at 60 °C for 12 h. The metakaolin used in the experiment is the product of kaolin calcined at a high temperature of 800 °C for 2 h, and its supplier is Chaopai Company in Inner Mongolia. The main components of metakaolin are SiO_2_ and Al_2_O_3_, of which the mass fraction of SiO_2_ is 56.91% and the mass fraction of Al_2_O_3_ is 42.35%.

Various methods were used to characterize the catalyst. Under a voltage of 10 kV, a scanning electron microscope (SEM, Thermo Fisher Scientific, FEI Quattro S, Waltham, MA, USA) was used to observe the morphology of the catalyst.

X-ray diffractometer (XRD, Rigaku, MiniFlex 600, Kyoto, Japan) was used to investigate the change in the crystal form of the catalyst. A Cu-Kα target with a wavelength of 0.154 nm was used for testing at a working voltage of 40 kV and a working current of 15 mA. The scanning range of the test data was 0° to 80°, 2θ degrees, and the scanning speed was 5°/min. Continuous sampling was selected as the sampling method.

The pore-size distribution and specific surface area of the catalyst were analyzed on a Brunauer-Emmett-Teller analyzer (BET, Micromeritics, Gemini VII 2390, Norcross, GA, USA). The BET equation was used to calculate the surface area of the sample, and the Barret-Joyner-Halenda (BJH) equation was used to calculate the pore-size distribution.

The liquid product was qualitatively and quantitatively analyzed using a gas chromatography-mass spectrometer (GC-MS, Agilent Technologies, 7890A-5975C, Santa Clara, CA, USA). Chromatographic conditions: Chromatographic column is HP-5MS (60 m × 250 μm × 0.25 μm); He was selected as carrier gas and the flow rate was 1.5 mL/min; split ratio 20:1, sample injection volume 1 µL, inlet temperature 280 °C. The test conditions of the mass spectrometry were an EI source; electron energy 70 eV; ion source temperature 230 °C; quadrupole 150 °C; scan mode was Scan; scan mass range was 50–550 u; solvent delay 4.0 min.

Ammonia temperature-programmed desorption (NH_3_-TPD, Altamira Instruments, AMI-300lite, Pittsburgh, PA, USA) experiment was performed to obtain the acidic-site type and concentration information of the catalyst. A total of 100 mg of sample were activated in an Ar flow at 550 °C for 0.5 h and then cooled to 100 °C. A gaseous ammonia flow (8 vol% in He, 30 mL/min) was introduced for 40 min to saturate the sample adsorption. Then, the sample was flushed with Ar (30 mL/min) for 40 min. The sample was heated from 100 °C to 600 °C in a He flow (20 mL/min) at a rate of 10 °C/min; the absorbed NH_3_ started to desorb from the sample, and the desorbed NH_3_ was monitored using a thermal conductivity detector.

Fourier transform infrared spectroscopy (FT-IR, Thermo Nicolet, iS50, Waltham, MA, USA) was performed in order to obtain the changes in the hydrocarbon composition of the liquid-phase products. The prepared samples were dried in a blast drying oven at 60 °C for 12 h, mixed with KBr powder and ground them into powder, then made it into tablets and tested in the range of 400–4000 cm^−1^.

In order to investigate the effect of the amount of alkali activator on the properties of the microspheres, metakaolin was used as the raw material; the alkali activator was sodium hydroxide; the molar ratio of H_2_O/Na_2_O was 20; the molar ratios of Na_2_O/SiO_2_ were 0.2, 0.4, 0.8, and 1.2, respectively. The dosages of sodium oxide were 1.36 g, 2.72 g, 5.43 g, and 8.15 g, respectively. A total of 1 g of microspheres were taken and put in a 100 mL beaker, and 80 mL of deionized water was added. The beaker was sealed with plastic wrap and placed in a constant-temperature water bath; the water-bath temperature was 80 °C and the water-bath time was 3 d. The prepared microspheres were tested by XRD, and the optimal dosage of the alkali activator was determined according to the conversion of the molecular sieve of microspheres.

In order to explore the effect of hydrothermal modification time on the properties of microspheres, NMGM catalysts were prepared under the optimal dosage of alkali activator. Under the determined optimal water-bath temperature, the hydrothermal conversion times were 0.5 d, 1.0 d, 1.5 d, 2.0 d, and 2.5 d, respectively. The specific surface area and crystal-form changes of NMGM prepared with different hydrothermal times were compared.

### 3.3. Reaction Mechanism of NMGM Thermal Catalytic Cracking of Waste LDPE

At present, the plastic-cracking mechanisms that have been widely recognized are mainly the free-chain scission mechanism and the carbocation mechanism [46]. The thermal catalytic-cracking method adopted in this work mainly follows the free-chain scission mechanism, and the catalytic-cracking process mainly follows the carbocation mechanism. The cleavage of plastic was divided into three stages: chain initiation, propagation stage, and chain termination (Figure 8). In the chain-initiation step, the free chains of long-chain LDPE molecules were cleaved into macromolecular fragments with radicals at one or both ends. The macromolecular LDPE fragments underwent a hydrogen-transfer reaction with the SiO_2_-Al_2_O_3_ framework to generate smaller LDPE fragments with carbocations. In the propagation stage of step 2, the free radical-bearing LDPE fragments are brought into contact with acidic sites on the surface and in the pores of the NMGM catalyst. In this step, the LDPE fragments tend to undergo a hydrogen transfer reaction, thereby abstracting hydrogen atoms from the LDPE macromolecular chain to restore the structure.

According to the compounds obtained in this work, it was inferred that the chain termination step in step three can be divided into four categories. NMGM were less acidic and therefore dominated by free-chain scission, in which a large number of LDPE-cracked fragments generated more stable alkanes through hydrogenation ((3-4) in Figure 8). The alkane products further reacted with the abundant hydroxyl groups on the surface of NMGM to generate a large number of alcohols ((3-3) in Figure 8). Fragments containing free radicals in the reaction underwent a hydrogen-transfer reaction to generate alkenes ((3-2) in Figure 8). Conjugated dienes reacted with olefins to undergo a Diels-Alder reaction to form cycloalkenes, and cycloalkenes further underwent a hydrogen-transfer reaction to form aromatic hydrocarbons ((3-1) in Figure 8) [47].

## 4. Conclusions

In this work, LDPE waste rich in municipal solid-plastic waste was catalytically pyrolyzed using a low-Si/Al-ratio NaA molecular-sieve NMGM catalyst with a SiO_2_/Al_2_O_3_ molar ratio of 1. NMGM was transformed into NaA molecular-sieve microspheres in a water bath at 80 °C under normal pressure for 1 d. The acid-enhanced geopolymer-based NaA molecular-sieve porous microspheres NMGM after hydrothermal treatment were confirmed to have potential significance in the thermal catalytic cracking of solid-plastic waste LDPE. The results of this study confirmed this idea, as NMGM catalyst actively participated in the thermal catalytic cracking of waste LDPE into fuel oil, and its liquid-phase yield reached 88.45% at 400 °C for 1 h. The total content of gasoline and diesel components accounted for 91.10% of the liquid product after the reaction. NMGM exhibits high catalytic activity and high selectivity for cracking waste LDPE into fuel oil. Given these facts, NMGM is considered as a promising catalyst for the degradation of municipal solid-waste plastics.

## Figures and Tables

**Figure 1 molecules-27-02557-f001:**
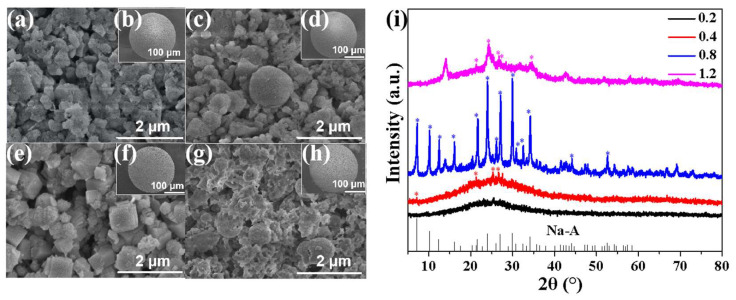
SEM and XRD tests of NMGM with different NaOH additions: (**a**,**c**,**e**,**g**) are the SEM tests of NMGM with Na_2_O/SiO_2_ of 0.2, 0.4, 0.8, 1.2, respectively; (**b**,**d**,**f**,**h**) SEM of NMGM with Na_2_O/SiO_2_ of 0.2, 0.4, 0.8, 1.2, respectively; (**i**) is the XRD pattern of different Na_2_O/SiO_2_ ratios (*: characteristic peaks of NaA molecular sieves).

**Figure 2 molecules-27-02557-f002:**
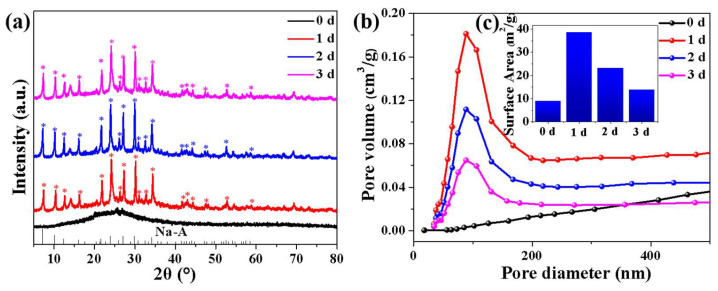
XRD test and BET test of NMGM catalyst with Na_2_O/SiO_2_ of 0.8 at hydrothermal time of 0 d, 1 d, 2 d, 3 d: (**a**) XRD (*: characteristic peaks of NaA molecular sieves), (**b**) pore-size distribution, (**c**) specific surface area.

**Figure 3 molecules-27-02557-f003:**
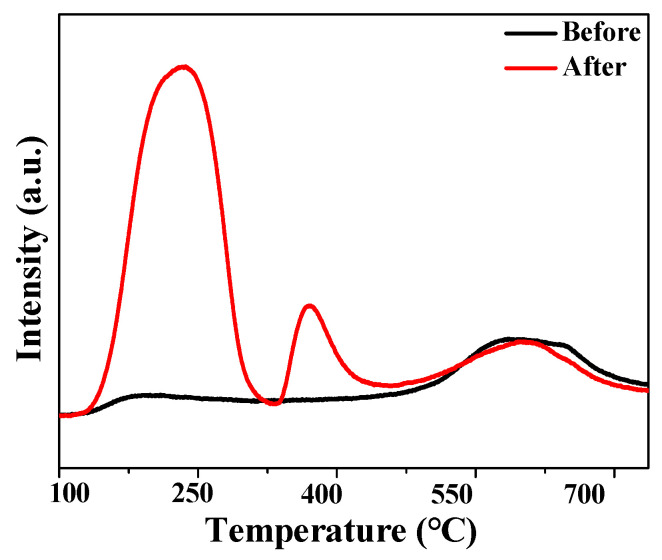
NH_3_-TPD tests of NMGM before and after hydrothermal 1 d.

**Figure 4 molecules-27-02557-f004:**
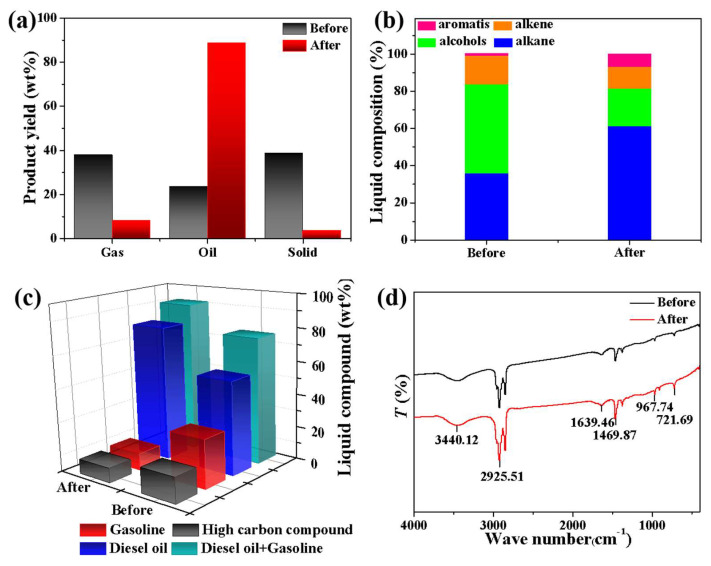
Product test of NMGM before and after hydrothermal thermal catalytic cracking of waste LDPE for 1 h (**a**) yield, (**b**) hydrocarbon composition distribution of liquid product, (**c**) liquid product composition distribution, (**d**) FTIR test.

**Figure 5 molecules-27-02557-f005:**
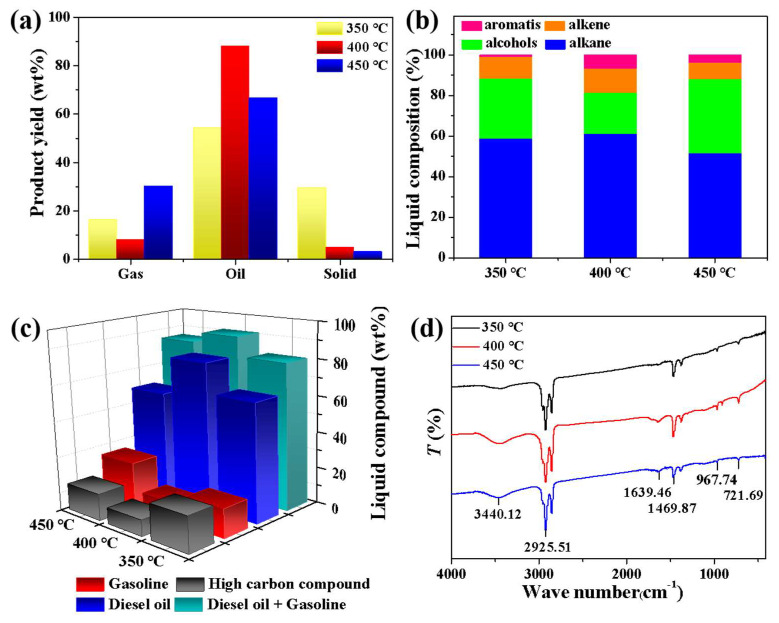
Product tests after thermal catalytic cracking of waste LDPE with NMGM at different reaction temperatures for 1 h (**a**) yield, (**b**) hydrocarbon composition distribution of liquid products, (**c**) liquid product-composition distribution, (**d**) FTIR test.

**Figure 6 molecules-27-02557-f006:**
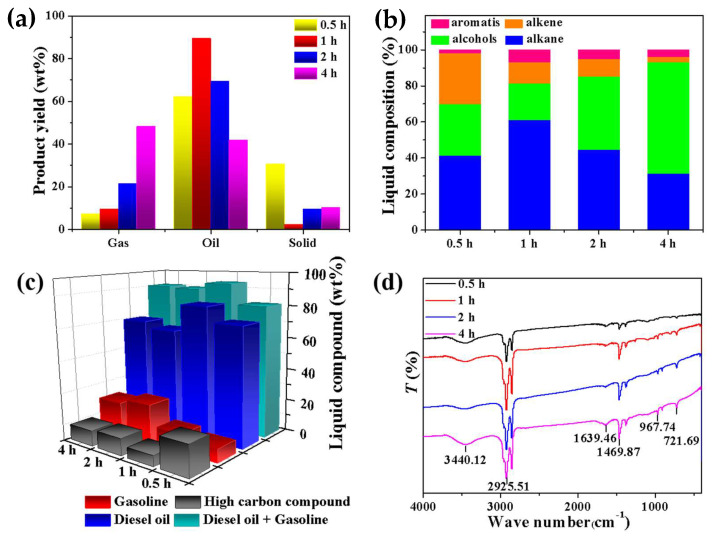
Product tests of NMGM thermal catalytic cracking of waste LDPE at 400 °C after different reaction times: (**a**) yield, (**b**) hydrocarbon-composition distribution of liquid products, (**c**) liquid product-composition distribution, (**d**) FTIR test.

**Figure 7 molecules-27-02557-f007:**
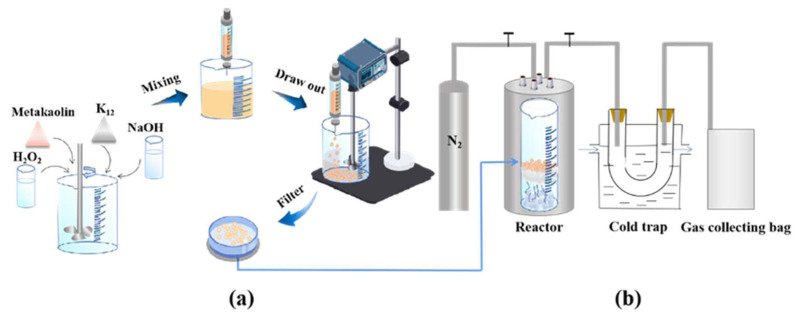
Flow chart of NMGM thermal catalytic cracking of waste LDPE: (**a**) Microsphere preparation process, (**b**) NMGM thermal catalytic cracking of LDPE waste.

**Figure 8 molecules-27-02557-f008:**
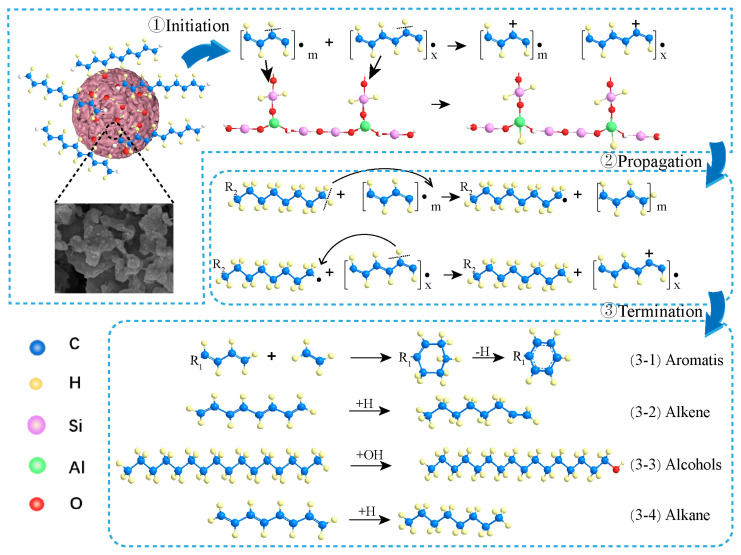
Image of the NMGM thermal catalytic-cracking LDPE mechanism.

## Data Availability

Not applicable.

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
