# Peer review of "Thermal Catalytic-Cracking Low-Density Polyethylene Waste by Metakaolin-Based Geopolymer NaA Microsphere"

_molecules, 2022, doi:10.3390/molecules27082557_

Round 1

Reviewer 1 Report

Manuscript ID: molecules-1671791
Title: Catalytic pyrolysis low density polyethylene waste by metakaolin-based geopolymer NaA microsphere

Reviewers' comments:

This study is focused on Catalytic pyrolysis low density polyethylene waste by me- 2 takaolin-based geopolymer NaA microsphere. The article contains detailed results which are both interesting and relevant to the journal. However, there are some issues that authors need to improve before this paper is accepted for publication. I would recommend the publication if authors make the following corrections based on my detailed comments as below.

Introduction is not very impressive. Authors have not gone through a detailed literature review on the subject. There are a lot of studies in 2019-2022 showing the effect of various catalyst on product yield and kinetic parameters and for genuine scientific reasons, authors should address the significance of this work especially with the work carried out most recently and why they were motivated by using this catalyst for degradation studies in presence of other commercial and synthesized catalysts i.e.,

i) "Pyrolysis of polypropylene over a LZ-Y52 molecular sieve: Kinetics and the product distribution." Iranian Polymer Journal28, no. 10 (2019): 839-847

ii) Cobalt-doped molecular sieve for efficient degradation of polypropylene into fuel oil: Kinetics and fuel properties of the oil." Chemical Engineering Research and Design177 (2022): 751-758.

iii) "Production of fuel oil and combustible gases from pyrolysis of polystyrene waste: Kinetics and thermodynamics interpretation." Environmental Technology & Innovation 24 (2021): 101996.

I have the following corrections and comments:

  1. The results and discussion is ahead of Materials and method. Is this the journal style, if not then rearrange them vice versa
  2. Lines 214-216, with increase in temperature, there is increase in gas fraction, then how the authors got maximum liquid yield.
  3. Line 218, Please replace “Onwudili JA” with “Onwudili”
  4. Line 222, please provide a GCMS chromatogram of the liquid fraction.
  5. Line 300, please replace “ml” with “mL”.
  6. Line 302, which type of thermocouple you have used, please name make and model, was it p-type or k-type
  7. Line 310, please name make and model of aluminum foil gas collection bag, you can also use a tedlar bag for it which is more secured
  8. The conclusion should be improved. The conclusion needs to highlight the applications of this study.

In general, the paper is well structured. The subject meets the aims and scope of the journal and it is of interest. I recommend the paper is accepted for publication in the Journal subject to the above-mentioned amendments

Author Response

Reviewer #1: This study is focused on Catalytic pyrolysis low density polyethylene waste by me- 2 takaolin-based geopolymer NaA microsphere. The article contains detailed results which are both interesting and relevant to the journal. However, there are some issues that authors need to improve before this paper is accepted for publication. I would recommend the publication if authors make the following corrections based on my detailed comments as below.

  1. Introduction is not very impressive. Authors have not gone through a detailed literature review on the subject. There are a lot of studies in 2019-2022 showing the effect of various catalyst on product yield and kinetic parameters and for genuine scientific reasons, authors should address the significance of this work especially with the work carried out most recently and why they were motivated by using this catalyst for degradation studies in presence of other commercial and synthesized catalysts i.e.
  2. i) "Pyrolysis of polypropylene over a LZ-Y52 molecular sieve: Kinetics and the product distribution." Iranian Polymer Journal28, no. 10 (2019): 839-847
  3. ii) Cobalt-doped molecular sieve for efficient degradation of polypropylene into fuel oil: Kinetics and fuel pro perties of the oil." Chemical Engineering Research and Design177 (2022): 751-758.

iii) "Production of fuel oil and combustible gases from pyrolysis of polystyrene waste: Kinetics and thermodynamics interpretation." Environmental Technology & Innovation 24 (2021): 101996.

Response: Thank you very much for your suggestion! the introduction of this manuscript was not written for the impressive purpose due to my lack of literature reading. The study of catalyst-to-product yield and kinetic parameters is crucial for pyrolysis of plastics. I have carefully consulted your comments and carefully read the literature you provided, and have expanded and cited the content at lines 88-107 of the manuscript.

  1. The results and discussion is ahead of Materials and method. Is this the journal style, if not then rearrange them vice versa.

Response: Thanks for your suggestion! Papers are formatted in strict accordance with Molecules' journal's Instructions for Authors, where Results and Discussion take precedence over Materials and Methods. The following two Molecules journal articles are examples of the typesetting I use for reference articles [1,2].

[1] M. Popova, Á. Szegedi, M. Oykova, H. Lazarova, N. Koseva, M.R. Mihályi, D. Karashanova, Y. Mitrev, P. Shestakova, Hydrodemethoxylation/Dealkylation on Bifunctional Nanosized Zeolite Beta, Molecules, 26 (2021).10.3390/molecules26247694

[2] T. Todorova, P. Petrova, Y. Kalvachev, Catalytic Oxidation of CO and Benzene over Metal Nanoparticles Loaded on Hierarchical MFI Zeolite, Molecules, 26 (2021).10.3390/molecules26195893

3.Lines 214-216, with increase in temperature, there is increase in gas fraction, then how the authors got maximum liquid yield.

Response: Thanks for your very professional advice! Lines 320-323 explained this. Onwudili et al. [3] found that LDPE thermally degrades to oil at 425 °C, and the reaction temperature exceeds this temperature, the proportion of oil product decreases due to its conversion to carbon and hydrocarbon gas. As the temperature increases, gas phase production The premise is that the activity of the catalyst is relatively stable. However, the carbon deposits generated by the reaction temperature are too high to block the pores, hinder the contact between the small molecular fragments after LDPE cracking and the reaction active sites, and reduce the catalytic activity of the catalyst. Therefore, there will be a more suitable reaction temperature for the preparation of oil products, and the gas-phase product yield will not increase indefinitely with the increase of the reaction temperature.

[3] Onwudili, J. A.; Insura, N.; Williams, P. T., Composition of products from the pyrolysis of polyethylene and polystyrene in a closed batch reactor: Effects of temperature and residence time. Journal of Analytical and Applied Pyrolysis 2009, 86, (2), 293-303.

  1. Line 218, Please replace “Onwudili JA” with “Onwudili

Response: Thanks for pointing this out! I've made the changes required at line 320.

  1. Line 222, please provide a GCMS chromatogram of the liquid fraction.

Response: Thanks for your suggestion! The hydrocarbon composition diagram in Fig. 5b and the distribution diagram of gasoline, diesel composition and high carbon composition according to the carbon number distribution in Fig. 5c are all obtained according to the statistics of GC-MS test data. Just plotting a GC-MS map doesn't give much more useful information. Therefore, it would seem repetitive to put GC-MS in the text. The GC-MS pattern in Figure 5 is shown below.

GC-MS tests of liquid product after thermal catalytic cracking of waste LDPE with NMGM at different reaction temperatures for 1 h

6.Line 300, please replace “ml” with “mL”.

Response: Thanks for your advice! I have made unit revisions and proofread throughout the text, which can be found on lines 408 and 416.

7.Line 302, which type of thermocouple you have used, please name make and model, was it p-type or k-type

Response: Thanks for your professional advice! The K-type WRNT-042 meter open probe thermocouple used in this article is ELECALL and is annotated in lines 410-411.

8.Line 310, please name make and model of aluminum foil gas collection bag, you can also use a tedlar bag for it which is more secured

Response: Thanks for your very professional advice! The foil airbag is a 3L foil airbag equipped with a single valve from Beekman Bio, identified on lines 419-420. Thanks for pointing out that Tedlar bags are safer, I will take this advice in future experiments.

9.The conclusion should be improved. The conclusion needs to highlight the applications of this study.

Response: Thanks for your advice! A discussion of the use of this study to produce fuel oil with high selectivity from MSW has been added to the conclusion, with all modifications found in lines 540-543.

Reviewer 2 Report

The use of waste materials as basis for chemicals and fuels production is of present interest, worldwide. In this paper, waste plastics catalytic conversion over metakaolin was proposed for oil production. In this respect, the manuscript is a contribution and potentially deserves publication. However, major revision is recommended in view of both the unclear obtained results and the need of further discussion. 

The following point in the introduction should be clarified, lines 70-73 “Catalytic pyrolysis is a one-stage reaction, and pyrolysis-catalytic cracking is a two-stage reaction. Compared with the two-stage reaction, catalytic pyrolysis can save the cost of experimental equipment. In this work, the catalytic pyrolysis method is adopted.” The authors mentioned that catalytic pyrolysis was used, however, the authors process physically separates pyrolysis from catalytic upgrading of pyrolysis volatiles. Moreover, the introduction should be improved. The authors should pay further attention to the technological aspects of the pyrolysis (one-stage) and pyrolysis-catalytic cracking (2 stages) strategies. For the reader, a short review of the state of the art and recent application in waste plastics catalytic pyrolysis technologies could be of great relevance. Thus, main reactor designs (and their combination in two stage process) application should be discussed, please at least consider fixed beds, fluidized beds, spouted beds, screw kilns…

The role played by hydrothermal treatment time on catalysts features should be further discussed and clarified.

The pyrolysis results reported in Figure 3b should not be reported together with TPD curves (Figure 3a). In the same line it is not appropriate to include these results in 2.1 section. Finally, these results were not commented in deep, the catalysts performance before and after hydrothermal treatment is uncertain. It could be expected lower gas yields for the catalysts with lower acidity. Please clarify.

According to the results reported in Figure 4a the authors obtained a complete conversion of plastic waste to volatiles (gas, oil and wax) operating at 350 ºC. This temperature seems to be too low to ensure complete polymer degradation, in fact higher temperatures were reported in previous literature to completely pyrolysis PE, please clarify.

The quality of Figure 4 c should be improved, it is difficult to read.

The following sentence is not really accurate, lines 214-216 “As the reaction temperature increased from 350 °C to 450 °C, the yield of gas phase products increased gradually, the yield of waxes decreased, and the yield of liquid oil products was the highest at 400 °C.” the authors should note that the minimum gas yield was obtained at 400 ºC.

The authors reported high contents of alcohols in the oil, however the polymer used in this study, PE, should not include oxygen. What is the source of this oxygen? Moreover, the authors stated that alcohols are stable compound; however, alcohols are highly reactive compounds.

The authors included a complete characterization of the catalysts prepared, unfortunately, this information was not used in the results discussion. This point should be improved.

A weak point of the paper is the lack of comparison with previous literature. The authors should provide a complete comparison of product yields and their composition with those reported in the literature with other technologies (fluidized beds, screw kilns, spouted beds, fixed beds, etc) and catalysts.

The selection of experimental conditions (temperature, catalysts/plastic ratio, residence time…) should be justified. Moreover, all the performed experiments should be described in detail in Materials and Methods section.

Author Response

Reviewer #2: The use of waste materials as basis for chemicals and fuels production is of present interest, worldwide. In this paper, waste plastics catalytic conversion over metakaolin was proposed for oil production. In this respect, the manuscript is a contribution and potentially deserves publication. However, major revision is recommended in view of both the unclear obtained results and the need of further discussion.

  1. lines 70-73 “Catalytic pyrolysis is a one-stage reaction, and pyrolysis-catalytic cracking is a two-stage reaction. Compared with the two-stage reaction, catalytic pyrolysis can save the cost of experimental equipment. In this work, the catalytic pyrolysis method is adopted.” The authors mentioned that catalytic pyrolysis was used, however, the authors process physically separates pyrolysis from catalytic upgrading of pyrolysis volatiles. Moreover, the introduction should be improved. The authors should pay further attention to the technological aspects of the pyrolysis (one-stage) and pyrolysis-catalytic cracking (2 stages) strategies.

Response: Thanks for your very professional advice! The one-step method and two-step method combining thermal cracking and catalytic cracking in the original text are biased. It has been revised and added as suggested at lines 75-87. In addition, all method names in the text have been corrected.

  1. For the reader, a short review of the state of the art and recent application in waste plastics catalytic pyrolysis technologies could be of great relevance. Thus, main reactor designs (and their combination in two stage process) application should be discussed, please at least consider fixed beds, fluidized beds, spouted beds, screw kilns…

Response: Thanks for pointing this out! It is necessary to review the latest progress and latest applications of waste plastic catalytic cracking technology, so according to your suggestion, I have added some reviews and discussions on fluidized bed, fixed bed and other reactors in lines 88-107, and some references are appropriately cited.

  1. The role played by hydrothermal treatment time on catalysts features should be further discussed and clarified.

Response:

Thanks for your very professional advice! The increase of the specific surface area helps the distribution of reactive sites and increases the reaction area, and the existence of mesopores can provide storage and reaction sites for free radicals and promote the further occurrence of cracking [4]. When the hydrothermal treatment time was 1 h, the specific surface area of the microspheres reached the maximum, and the amount of mesopores and macropores increased. The increase of the specific surface area and the existence of mesopores had a positive effect on the catalytic activity. The test results of catalyst pore size distribution and specific surface area at different hydrothermal times can be seen in Figure 2 in the text.

The discussion has been revised and added as suggested, and can be found in lines 274-295 of the text. All modifications have been marked in red in the mentioned lines.

 [4] Sakata, Y.; Azhar Uddin, M.; Muto, A.; Kanada, Y.; Koizumi, K.; Murata, K., Catalytic degradation of polyethylene into fuel oil over mesoporous silica (KFS-16) catalyst. Journal of Analytical and Applied Pyrolysis 1997, 43, (1), 15-25.

  1. The pyrolysis results reported in Figure 3b should not be reported together with TPD curves (Figure 3a). In the same line it is not appropriate to include these results in 2.1 section. Finally, these results were not commented in deep, the catalysts performance before and after hydrothermal treatment is uncertain.

Response: Thanks for your suggestion! As you suggested, the pyrolysis results were separated from the TPD test results. A detailed discussion of the effect of hydrothermal treatment before and after hydrothermal treatment on the pyrolysis waste polyethylene is added, and Figure 4 is added. All modifications are marked in lines 274-295.

5.It could be expected lower gas yields for the catalysts with lower acidity. Please clarify.

Response: Thanks for your suggestion! Catalysts with strong acid sites will promote the occurrence of secondary reactions and tend to generate more gas-phase products and very few liquid-phase oil products. Lines 255-259 add bibliographic support and explain this.

  1. According to the results reported in Figure 4a the authors obtained a complete conversion of plastic waste to volatiles (gas, oil and wax) operating at 350 ºC. This temperature seems to be too low to ensure complete polymer degradation, in fact higher temperatures were reported in previous literature to completely pyrolysis PE, please clarify.

Response: Thanks for your very professional advice! In this experiment, wax was defined as the mass of the solid product remaining in the reactor in the product after the reaction. It contained incompletely reacted raw materials and waxy products that have undergone preliminary cracking. So the sum of gas, liquid oil and solid matter products would reach 100%. It was inappropriate to refer to it collectively as the quality of the wax, as you suggested, modify the original wax yield to the solid product yield. In Figure 4, Figure 5, Figure 6 and lines 313, 434, and 438 in the text have been modified.

7.The quality of Figure 4 c should be improved, it is difficult to read.

Response: Thanks for your advice! The label font and coordinate direction in Figure 5c pointed out in the suggestion have been adjusted, and Figures 4c and 6c have also been changed for a unified format and convenient reference. After checking the full text images, the font size of the title and coordinates has been increased in Figure 2c.

8.The following sentence is not really accurate, lines 214-216 “As the reaction temperature increased from 350 °C to 450 °C, the yield of gas phase products increased gradually, the yield of waxes decreased, and the yield of liquid oil products was the highest at 400 °C.” the authors should note that the minimum gas yield was obtained at 400 ºC.

Response: Thanks for your professional advice! The yield of gas phase components is affected by temperature, but not linearly. A more precise discussion has been made as you suggested, the modifications can be seen in lines 313-316.

9.The authors reported high contents of alcohols in the oil, however the polymer used in this study, PE, should not include oxygen. What is the source of this oxygen? Moreover, the authors stated that alcohols are stable compound; however, alcohols are highly reactive compounds.

Response: Thanks for your very professional advice! The in-situ transformation of metakaolin-based geopolymers into Na-A molecular sieves is inseparable from the effect of alkali activators. In lines 159-191, the amount of alkali activators to the degree of molecular sieve conversion of NMGM microspheres is discussed. When Na2O/SiO2 is 0.8, per 10 g of metakaolin, the amount of sodium hydroxide added is 5.43 g, and the amount of sodium hydroxide is a lot. Although the hydrothermal treatment was carried out in subsequent work, abundant hydroxyl groups remained on the surface of the microspheres and inside the pores, and NMGM was a catalyst with both acidic (bronsted and Lewis) and basic properties. Therefore, in the presence of abundant hydroxyl groups, alcohols in the product can exist relatively stably. The formation mechanism of alcohols is discussed in subsection 3.3, pp. 524-525.

10.The authors included a complete characterization of the catalysts prepared, unfortunately, this information was not used in the results discussion. This point should be improved.

Response: Thanks for your very professional advice! In this paper, we hope to investigate the catalytic cracking reaction of Na-A molecular sieve geopolymer microspheres on waste LDPE. Therefore, by determining the optimal amount of alkali activator, the Na-A molecular sieve geopolymer microspheres with the most complete crystal form are prepared. The increase of specific surface area and the existence of mesopores have a positive effect on the catalytic activity of plastic cracking catalysts. Through the investigation of hydrothermal time, the NMGM with the most perfect crystal structure and the largest specific surface area and average pore size was prepared. In the results and discussion, a discussion of the thermal catalytic cracking of waste LDPE with NMGM microspheres before and after hydrothermal has been added, which can be found in lines 274-295 of the text.

11.A weak point of the paper is the lack of comparison with previous literature. The authors should provide a complete comparison of product yields and their composition with those reported in the literature with other technologies (fluidized beds, screw kilns, spouted beds, fixed beds, etc) and catalysts.

Response: Thanks for your very professional advice! In this paper, a comparison between the catalytic technology using fluidized bed, fixed bed and other reactors and this work is added. The cracking temperatures here are lower and higher liquid yields are achieved. A detailed discussion can be found in lines 88-107.

12.The selection of experimental conditions (temperature, catalysts/plastic ratio, residence time…) should be justified. Moreover, all the performed experiments should be described in detail in Materials and Methods section.

Response: Thanks for your advice! The experimental conditions such as the experimental temperature, the ratio of catalyst to plastic, and the residence time were determined on the premise of referring to many literatures [5-8]. In the experimental part, the specific experimental steps of the hydrothermal time of the microspheres and the dosage of the alkali activator are added, which can be found in lines 482-497 of the text.

 [5] Zhang, G.; Zhang, X.; Bai, T.; Chen, T.; Fan, W., Coking kinetics and influence of reaction-regeneration on acidity, activity and deactivation of Zn/HZSM-5 catalyst during methanol aromatization. Journal of Energy Chemistry 2015, 24, (1), 108-118.

[6] Aguado, J.; Serrano, D. P.; Escola, J. M., Fuels from Waste Plastics by Thermal and Catalytic Processes: A Review. Industrial & Engineering Chemistry Research 2008, 47, (21), 7982-7992.

[7] Arshad, H.; Sulaiman, S. A.; Hussain, Z.; Naz, M. Y.; Moni, M. N. Z., Effect of Input Power and Process Time on Conversion of Pure and Mixed Plastics into Fuels Through Microwave-Metal Interaction Pyrolysis. Waste and Biomass Valorization 2021, 12, (6), 3443-3457.

[8] Fivga, A.; Dimitriou, I., Pyrolysis of plastic waste for production of heavy fuel substitute: A techno-economic assessment. Energy 2018, 149, 865-874.

Round 2

Reviewer 1 Report

Minor revision: please avoid full name of authors in text citation

Reviewer 2 Report

The authors have revised and improved the paper, moreover the questions and doubts raised by the reviewer were suitably clarified. Accordingly, the paper deserves publication as it stands.